# The Positive Association of Plasma Levels of Vitamin C and Inverse Association of VCAM-1 and Total Adiponectin with Bone Mineral Density in Subjects with Diabetes

**DOI:** 10.3390/nu14193893

**Published:** 2022-09-20

**Authors:** Sushil K. Jain, William E. McLean, Christopher M. Stevens, Richa Dhawan

**Affiliations:** Departments of Pediatrics and Medicine, Louisiana State University Health Sciences Center-Shreveport, 1501 Kings Highway, Shreveport, LA 71103, USA

**Keywords:** vitamin C, bone mineral density, type 2 diabetes, VCAM-1, inflammation, adiponectin

## Abstract

Context. Population studies have shown a trend in decreasing vitamin C status and increasing prevalence of osteoporosis in patients with diabetes and non-diabetic people. Dietary vitamin C consumption is linked to improvement in bone mineral density (BMD) in epidemiological studies. VCAM-1 and adiponectin are known to activate osteoclasts, which increase bone loss. Aim: This study examined whether there is any association between the circulating level of vitamin C and BMD and whether the beneficial effect of vitamin C on BMD was linked to a simultaneous decrease in circulating levels of adiponectin and VCAM-1 in subjects with diabetes. Methods: Patients with diabetes (T2D, *n* = 74) and age-matched non-diabetic controls (*n* = 26) were enrolled in this study. Fasting blood levels of glycemia, adiponectin, VCAM-1, inflammation biomarkers, and vitamin C were determined in both groups. The BMD of the lumbar spine (L1–L4), left femur, and right femur was determined using a DXA scan in subjects with diabetes. Results: Patients with diabetes had lower levels of vitamin C and higher levels of VCAM-1 and inflammatory cytokines. There was a significant positive association between vitamin C blood levels and lumbar spine BMD as well as a significant negative association between total adiponectin and VCAM-1 levels with that of vitamin C and lumbar BMD in patients with diabetes. Total adiponectin and VCAM-1 also showed a negative association with BMD of both the right and left femurs. The inter-relationship among the circulating levels of vitamin C and VCAM-1 and BMD was strong and is a novel finding. Conclusions: This study reports a positive association of circulating vitamin C levels and the BMD and that the beneficial effects of vitamin C on BMD could be linked to a simultaneous lowering in circulating VCAM-1 and total adiponectin levels. Thus, dietary vitamin C consumption has potential to lower inflammation and the risk of osteoporosis in subjects with diabetes.

## 1. Introduction

Bone mineral density (BMD) is frequently used as a predictor of bone health and to measure the risk of osteoporosis [1,2]. Regulation of BMD is dependent on extrinsic environmental lifestyle and intrinsic factors called genetics. Among the environmental factors, nutrition has been shown to influence peak bone mass and the modulation of bone health [1,2]. Poor bone health in subjects with diabetes could vary based upon factors such as immune and vascular abnormalities and musculoskeletal changes due to poor hyperglycemia control [3,4].

Vitamin C (ascorbic acid) is an essential nutrient [5,6,7]. Vitamin C is a coenzyme for prolyl and lysyl hydroxylases, enzymes essential for proline hydroxylation in collagen biosynthesis and the differentiation of the cells (osteoblasts) required for optimal bone mineral density and bone health [5,6,8]. Vitamin C deficiency has been shown to cause disordered bone formation in a rat model [7]. Malabsorption of vitamin C has been linked to the development of low BMD and osteoporosis in inflammatory bowel disease patients [9]. Scholarly reviews conclude that consumption of a vitamin-C-rich diet was negatively associated with decreased risk of hip fracture, osteoporosis, and loss of BMD [10,11,12,13,14]. Rondanelli et al. [14] elegantly summarized various studies providing evidence for a positive association between vitamin C supplementation and BMD improvement. A few studies in elderly subjects reported lower serum vitamin C levels in hip fracture patients [15,16]. Studies investigating the association between circulating levels of vitamin C and BMD are limited.

Osteoclasts and osteoblasts are the cells primarily responsible for the breakdown and formation of bone, respectively [17]. Inflammatory cytokines are known to induce osteoclast differentiation [18], and multiple studies have shown a negative correlation between BMD and circulating levels of inflammatory biomarkers [19,20]. Vascular cell adhesion molecule-1 (VCAM-1), an adhesion protein known for its role in the inflammatory response [21], has shown to elevate osteoclast activity by recruiting monocytic osteoclast progenitors, and antibodies against VCAM-1 have been shown to be beneficial in preserving bone structure [22]. No previous study has investigated whether circulating vitamin C levels negatively influence VCAM-1 levels, which in turn have a positive association with BMD.

Adiponectin is secreted by adipocytes and adiponectin receptors present in primary human osteoblasts from the femur and tibia [23]. Adiponectin stimulates the osteoclast bone-resorbing RANK pathway [24,25]. Clinical studies report a negative correlation between circulating total adiponectin levels and the BMD of various skeletal sites [26,27,28,29,30,31].

This study examined the effect of vitamin C status on VCAM-1, total circulating levels of adiponectin, and BMD and whether the levels of vitamin C, adiponectin, VCAM-1, and BMD are interrelated. This study demonstrates a positive association of vitamin C and an inverse association of total adiponectin and VCAM-1 plasma levels with BMD in patients with diabetes and that the beneficial effects of vitamin C on BMD could be linked to a simultaneous decrease in circulating levels of VCAM-1 and total adiponectin.

## 2. Materials and Methods

Enrollment of Subjects: Written informed consent obtained from all subjects with diabetes and non-diabetic subjects was, as per protocol, approved by the Institutional Review Board for Human Experimentation (IRB Protocol # H-09-073). All participants included in this study were adult type 2 diabetics (T2D), and subjects who agreed to participate were consecutively enrolled in this study. Fasting blood was collected. Of the 100 enrolled subjects, 74 subjects came to the clinic for blood draw and measurement of BMD. Non-diabetic subjects were mostly relatives or friends of the diabetic patients. BMD was not determined in the non-diabetic subjects because this study focus was not to compare the BMD levels between the groups.

Inclusion/Exclusion Criteria: Adult subjects with diabetes and non-diabetic subjects aged 25–65 years were included in the study. Patients with any history of cardiovascular disease, sickle cell disease, insulin intake, or metabolic disorders, including uncontrolled hypertension, hypothyroidism, or hyperthyroidism, were excluded, including those who showed any signs of hepatic dysfunction, which is defined as any underlying chronic liver disease or liver function tests greater than 1.5 times normal, or renal dysfunction, defined as creatinine greater than 1.5 mg/dL. This study did not include total blood testosterone levels in men as an exclusion criterion [32,33,34]. Women with a positive pregnancy test or those nursing infants were excluded as well. Subjects taking any herbal or vitamin supplement were also excluded. Similar inclusion/exclusion criteria were used for enrolling non-diabetic, healthy subjects.

Blood collection: Blood was collected after an overnight fast (8 h). Following blood collection, serum tubes for chemistry profile, EDTA tubes for HbA1C, and complete blood counts were delivered to the LSUHSC clinical laboratories. Extra EDTA-blood tubes were brought to the research laboratory. Clear plasma was isolated via centrifugation at 3000 rpm (1500× *g*) for 15 min. Fresh plasma was used for VC determinations. The remaining plasma samples were stored at −80 °C for analyses of inflammatory cytokine parameters.

VCAM-1, total adiponectin, and inflammation biomarkers assays: Plasma levels of total adiponectin, VCAM-1, IL-8, TNF-α, and IL-1β were determined using the sandwich ELISA method with kits purchased from ThermoFisher Scientific Co. (Rockford, IL, USA). All appropriate controls and standards as specified by the manufacturer’s kit were used. In the cytokine assay, control samples were analyzed each time to check the plate-to-plate variation on different days of analysis [35].

Vitamin C and bone mineral density (BMD) assays: Vitamin C was determined in fresh plasma using the metaphosphoric acid and dinitrophenylhydrazine method described by Park et al. [36]. BMD was measured, as described previously [37], at the lumbar spine (L1–L4) and proximal femur (hip) on both sides by DXA using a LUNAR DPX densitometer (GE-LUNAR, Madison, WI, USA). All measurements were obtained and analyzed using standard protocols provided by the manufacturer. BMD values are expressed as g/cm^2^.

All chemicals were obtained from Sigma Chemical Co. (St. Louis, MO, USA) unless otherwise indicated. Data were analyzed statistically using one-way ANOVA among different groups using Sigma Plot 14.5 software (Jandel Scientific, San Rafael, CA, USA). Correlation coefficient r values were obtained by using the Pearson or Spearman correlation. *p* values were determined using multiple regression analyses and controlling for BMI and duration of diabetes. A *p*-value ≤ 0.05 was considered significant.

## 3. Results

Table 1 shows the ages, BMI, and number of subjects in the T2D and non-diabetic control groups. Table 1 also shows the glucose, glycated hemoglobin, total adiponectin, and VCAM-1 levels of subjects in both groups and the BMD of the T2D patients. The ages in the control and T2D groups were similar. BMI was significantly higher in the diabetic group than in the control group. The levels of adhesion molecules (VCAM-1) and pro-inflammatory cytokines (IL-8, TNF-α, IL-1β) were significantly elevated in the diabetic group. Total adiponectin levels were similar in both groups. There was an increase in the triglyceride/HDL cholesterol ratio in subjects with diabetes, but there was no difference in the LDL/HDL cholesterol ratio between the two groups. Lumbar spine, right femur, and left femur BMD levels are given for subjects in the diabetic group, but BMD was not determined in the non-diabetic control subjects.

The relationship between the lumbar spine BMD and the levels of VC (r = 0.30, *p* = 0.01), total adiponectin (r = −0.36, *p* = 0.01), and VCAM-1 (r = −0.40, *p* = 0.01) in subjects with diabetes is given in Figure 1. This shows that vitamin C had a significant positive association with lumbar spine BMD, whereas both total adiponectin and VCAM-1 showed a negative association with the lumbar spine BMD. 

Figure 2 illustrates that while the left femur BMD had a negative association with VCAM-1 (r = 0.22, *p* = 0.04) and total adiponectin (r= −0.30, *p* = 0.01) levels, it showed no significant correlation with VC. Similarly, the right femur BMD showed an inverse association with VCAM-1 (r= −0.25, *p* = 0.03) and total adiponectin (r= −0.41, *p* = 0.01) but showed no significant association with vitamin C (Figure 3). The relationship was determined after controlling for BMI, duration of diabetes, and age. It appears that the lumbar spine BMD had the most positive response to the circulating vitamin C status.

Figure 4 shows a significant correlation between the blood levels of vitamin C and VCAM-1 (r= −0.30, *p* = 0.01) and total adiponectin (r= −0.22, *p* = 0.05) in T2D patients. There was a simultaneous lowering of VCAM-1 and total adiponectin with increase in vitamin C levels in the blood of subjects with diabetes.

HbA1c levels did not demonstrate any relationship with the vitamin C level or with the BMD of the lumbar spine or the right or left femur. Similarly, there was no relationship of the LDL/HDL cholesterol ratio or the triglyceride/HDL cholesterol ratio with the BMD or vitamin C levels. The vitamin C levels showed an inverse association with TNF-α and IL-1β levels but not with IL-8 levels. However, no association was seen between TNF-α or IL-1β with the BMD of the lumbar spine or either femur. IL-8 did show a negative association with BMD in the lumbar spine but not with either femur. We did not observe any significant interrelationship between BMD and circulating levels of TNF-α, IL-1β, or IL-8 and vitamin C (data not shown here).

## 4. Discussion

Many factors, including genetics, vitamin C, calcium, age, and hyperglycemia and body weight, are known to influence BMD [5,14,16]. Vitamin C positively upregulates expression of bone matrix genes in osteoblasts and increases trabecular bone formation [13,14]. Several studies carried out in human populations and using a genetic mouse model have shown that vitamin C exerts a positive effect on bone health [13,14]. Dietary consumption of vitamin C is required since the human body cannot synthesize or store vitamin C [38]. Citrus fruits, such as grapefruit and oranges, as well as strawberries, potatoes, tomatoes, peppers, cabbage, Brussels sprouts, broccoli, and spinach are rich sources of vitamin C. Vitamin C status has been shown to be decreasing in population studies [39], and unfortunately, vitamin C deficiency could become more common, especially in low-income groups and countries [40]. Survey data from the National Health and Nutrition Examination report an increase in the prevalence of osteoporosis in T2D and non-diabetic people from 2005–2006 to 2013–2014 [41].

The present study found that a significant association exists between the blood levels of vitamin C and the lumbar spine BMD in patients with diabetes. This finding is interesting because our study was carried out in younger subjects, unlike in previous studies carried out in older subjects [15,16]. The strong interrelationship reported here among the levels of vitamin C and VCAM-1 and BMD is novel. Vitamin C has a strong oxidoreduction property [5,38]. Adequate vitamin C status accelerates hydroxylation reactions by maintaining the active center of metal ions in a reduced state for the optimal activity of hydroxylase and oxygenase enzymes crucial in collagen formation and bone mineralization [5]. Low levels of vitamin C can result in decreased antioxidant capacity and thereby increase oxidative stress and activation of pro-inflammatory cytokines and VCAM-1 levels [42,43]. Similar to the data reported in some studies in the literature, we also observed a negative association between IL-8 and the lumbar spine BMD. VCAM-1 has been shown to increase osteoclast activity, with inhibition of VCAM-1 helping to preserve bone structure [22]. The interrelationship between vitamin C, VCAM-1, and BMD reported in the present study suggests that the beneficial effect of vitamin C on BMD could be linked to a simultaneous decrease in circulating VCAM-1 levels.

This study reports a significant association between total adiponectin and the BMD of the lumbar spine, right femur, and left femur. Previous studies have also reported an association between adiponectin and the lumbar spine but not the femurs, nor all three [28,29,30,31]. Another study reported that increasing levels of adiponectin and lower BMD levels were positively associated with osteocalcin, a biomarker of bone resorption [44]. Adiponectin stimulates the osteoclast RANKL pathway [24,25]. The present study reports a negative correlation of total adiponectin with vitamin C levels. This is a new finding and suggests that the positive association of vitamin C with BMD may be linked to lower levels of adiponectin and VCAM-1 and thereby downregulation of the bone resorption pathway.

Two major cell types, osteoclasts and osteoblasts, are primarily responsible for bone homeostasis. These two cell types work in opposition to one another, with osteoclasts responsible for breaking down bone tissue and osteoblasts responsible for bone formation [17]. Osteoblasts and osteoclasts interact with each other through receptor activation of the NF-kB ligand (RANKL), located on osteoblasts, and receptor activation of nuclear factor k-B (RANK), expressed on osteoclasts. Osteoclasts are activated when RANKL binds to RANK, resulting in increased bone breakdown [45]. A disruption in this interaction between RANKL and RANK can lead to increased osteoclast activity, resulting in decreased bone mineral density (BMD) and bone strength due to an increased rate of bone breakdown [46]. Chronic inflammation is a well-known risk factor for bone loss due mainly to overexpression of RANKL during inflammatory conditions [47]. During the inflammatory state, RANKL is expressed by activated T cells and B cells in addition to osteoblasts. This increases bone loss due to osteoclast hyperactivity via the RANK and RANKL interaction [48,49]. VCAM-1 has been shown to elevate osteoclast activity by recruiting monocytic osteoclast progenitors, and antibodies against VCAM-1 have been shown to be beneficial in preserving bone structure [22]. Wang et al. recently showed that suppressing VCAM-1 resulted in decreased osteoclast activity and increased BMD [50]. A previous study showed that there was a significant correlation between higher VCAM-1 expression and lower hip and spine BMD [48]. Stimulation of VCAM-1 by RANKL has also been reported [51]. The limitations of this study are the lack of BMD data on normal subjects and that total testosterone was not included as an exclusion criteria during enrollment of subjects. This study suggests that VCAM-1 is an important mediator of inflammation and can have a strong negative impact on the BMD. This study further shows that vitamin C has the potential to suppress circulating VCAM-1 and thereby protect against bone loss and reduced BMD.

## 5. Conclusions

Figure 5 summarizes the mechanisms of the positive effects of vitamin C on bone health, which could involve both the activation of the hydroxylation reactions required for collagen formation and bone mineralization as well as the antioxidant property of vitamin C that lowers inflammation, thus preventing RANKL and osteoclast activation and bone loss. Therefore, vitamin C can contribute beneficially to bone health through the potentiation of osteoblast differentiation and bone formation as well as the suppression of osteoclast hyperactivity. Better nutrition and dietary intake of vitamin C rich fruits and vegetables is required to decrease the risk and possibly prevent the increasing incidence of osteoporosis in our population. 

## Figures and Tables

**Figure 1 nutrients-14-03893-f001:**
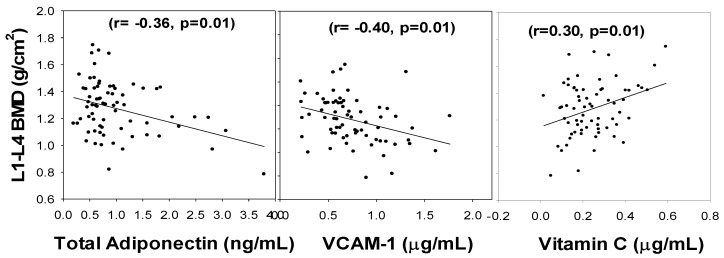
Correlation of Lumbar spine BMD with blood levels of total adiponectin, VCAM-1, and vitamin C in subjects with diabetes. *p*-values were determined using multiple regression analyses and controlling for BMI and duration of diabetes.

**Figure 2 nutrients-14-03893-f002:**
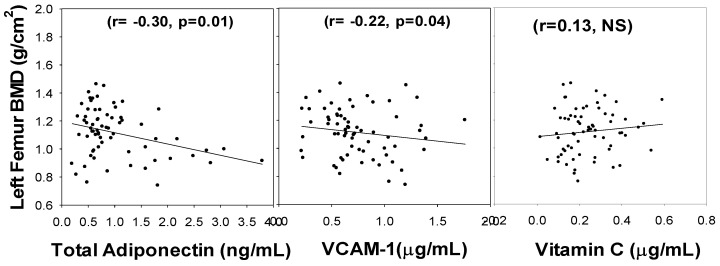
Correlation of Left Femur BMD with blood levels of total adiponectin, VCAM-1, and vitamin C in subjects with diabetes. NS, not-significant. *p*-values were determined using multiple regression analyses and controlling for BMI and duration of diabetes.

**Figure 3 nutrients-14-03893-f003:**
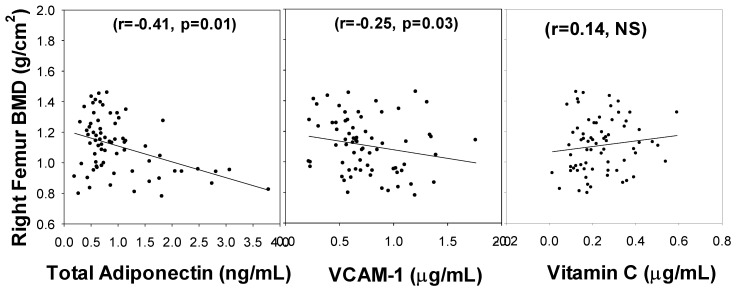
Correlation of Right Femur BMD with blood levels of total adiponectin, VCAM-1, and vitamin C in subjects with diabetes. NS, not-significant. *p*-values were determined using multiple regression analyses and controlling for BMI and duration of diabetes.

**Figure 4 nutrients-14-03893-f004:**
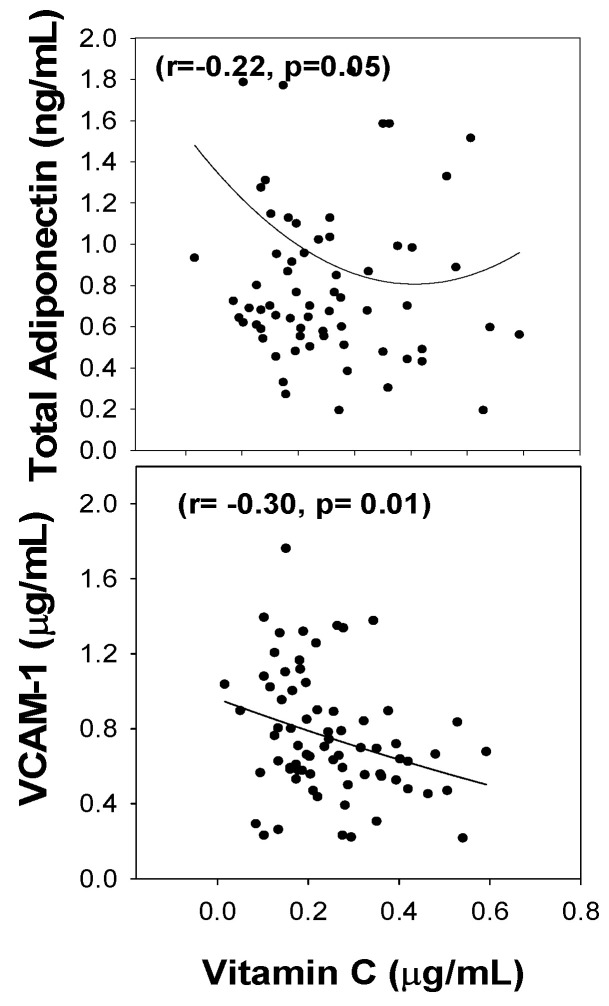
The association of total adiponectin and VCAM-1 with vitamin C blood levels in subjects with diabetes.

**Figure 5 nutrients-14-03893-f005:**
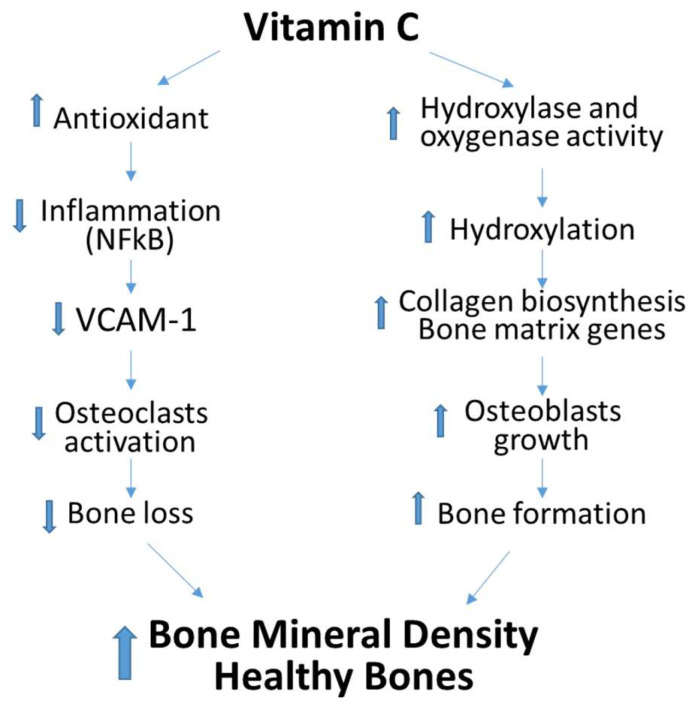
Proposed mechanism of positive effects of dietary vitamin C on Bone Mineral Density. Up-Arrow indicates upregulation or increase and down-Arrow indicates down regulation or decrease.

**Table 1 nutrients-14-03893-t001:** Ages, BMI, duration of diabetes, BMD, and various blood biomarkers in subjects with diabetes (T2D) and non-diabetic control subjects. Values (mean SE) marked * are significantly different (*p* < 0.05). ND, not determined.

	T2D	Non-Diabetic Controls
*n*	74	26
Age (years)	49.6 ± 1.1	46.6 ± 1.9
Gender (F/M)	54/20	20/6
BMI	36.5 ± 1.03	29.8 ± 1.3 *
Duration of diabetes (years)	4.2 ± 0.5	NA
HbA1C (%)	7.71 ± 0.20	ND
Glucose (mg/dL)	139 ± 6	93 ± 4 *
Calcium (mg/dL)	9.33 ± 0.05	9.98 ± 0.42
Lumbar (L1–L4) BMD (g/cm^2^)	1.27 ± 0.02	ND
Right femur BMD (g/cm^2^)	1.11 ± 0.02	ND
Left femur BMD (g/cm^2^)	1.12 ± 0.02	ND
Vitamin C (µg/dL)	0.25 ± 0.01	0.32 ± 0.03 *
Total adiponectin (ng/mL)	1.03 ± 0.08	1.09 ± 0.22
VCAM-1 (µg/mL)	0.80 ± 0.04	0.65 ± 0.06 *
IL-8 (pg/mL)	3.90 ± 0.23	2.59 ± 0.13 *
IL-1β (pg/mL)	10.5 ± 0.4	7.5 ± 1.02 *
TNF-α (pg/mL)	129.6 ± 12.1	68.1 ± 9.0 *
LDL/HDL-chol (ratio)	2.23 ± 0.11	2.12 ± 0.17
TG/HDL-chol (ratio)	4.73 ± 0.60	1.90 ± 0.25 *

## Data Availability

Data can be available if requested.

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
