# Peer review of "The Positive Association of Plasma Levels of Vitamin C and Inverse Association of VCAM-1 and Total Adiponectin with Bone Mineral Density in Subjects with Diabetes"

_nutrients, 2022, doi:10.3390/nu14193893_

Round 1
Reviewer 1 Report
This is an interesting study. Anyway, I have some comments and suggestions.
Title and Abstract
- please, replace in the text too "diabetic patients" in "patients with diabetes"
- You start with aim writing about background; please, divide better this section.
Introduction
- Please, about regulation of BMD could be complete to introduce more data, not only lifestyle and genetics. See PMID 26980458 and PMID 35101185.
- Clarify your aim in the text;
- This isn't the right section to insert your results!
Materials and methods
- have you calculated the sample size?
- the subjects were enrolled consecutively?
- Specify, how many males and females subjects
- please, clarify why your range of age enrollment was >25.
- please, specify why you did not evaluate BMD in control group.
- In exclusion criteria, have you checked total testosterone? see
- PMID: 33735389, PMID 29888533
- about the use of 2 different type of DXA, have you provided the statistical correction of these data?
Discussion
Please, insert the limitations of your study, possibly including the total testosterone check, the use of two different DXa, if not adjusted results, the absence of DXA in control group etc.
- Please, insert the future perspectives of your results.
Conclusions:
- Could be better to do not use references in your conclusions.
Author Response
All of the comments are addressed and are included in the revised manuscript.
Authors are indeed grateful to the reviewers for the time spent and for the excellent comments.
-----------------
Title, abstract and text has been changed from diabetic patients to ‘patients with diabetes’.
Aim has been divided into two subheadings, Context and Aim.
Introduction:
As advised by the reviewer introduction has been revised and suggested references included in the revised manuscript (lines 39-41).
Aim is clarified in the introduction section (lines 69-71).
Materials and Methods:
No, we did not calculate sample size, subjects were enrolled consecutively (lines 79-80), number of males and females in each group are included in Table 1, there was no specific justification for age range of subjects, BMD was not determined because our aim was not to compare the BMD between the two groups, total testosterone was not used as an exclusion criteria has been included in the revised text (lines 91-92), study used only one DXA, we apologize for this error in methods section.
Discussion: As suggested by the reviewer limitations of study are included in the discussion (lines 222-224)
References in conclusion section are deleted.
Reviewer 2 Report
The positive association of plasma levels of vitamin C, and in-2 verse association of VCAM-1 and total adiponectin, with bone 3 mineral density in Type 2 diabetic patients
Dear Authors,
Congratulations on writing such an interesting article. This manuscript aims to reveal the association of vitamin C, and an inverse association of total adiponectin and VCAM-1 plasma levels, with BMD in T2D patients.
The following are my comments and suggestions:
Abstract and Introduction:
1. My most of comments are on the presentation of information. The manuscript needs to present the pictorial presentation of the mechanism of action of VCAM-1 and Vit C. For example, kindly provide the diagrams for VCAM-1 and their role in the diabetic condition associated with bone disorder establishment.
2. There were several errors in spelling throughout the document, mainly because the hyphen was used in many places unnecessarily. Kindly fix it.
3. Kindly provide information about the quantity of male or female participants in this study. Any interesting/distinct observation between gender?
4. Does any candidate face a hyperosmolar hyperglycemic state?
5. Kindly provide the information about LDL/HDL ratio, TG/HDL-C, non-HDL/HDL, and Apo B/Apo A ratio and their relationship with VCAM-1 under diabetic conditions.
6. Kindly provide brief details about Dual-energy x-ray absorptiometry (DEXA) originated data such as bone mineral content (BMC), fat-free mass (FFM) studies and their association with VCAM-1 or other cytokines.
Discussion:
Kindly provide brief details about the action of the mechanism of VCAM-1 & Vit C and how they get modulated in case of Diabetic conditions. Kindly explain the case of what is the most interesting feature that makes Vit C a potential therapeutic agent.

Author Response
Authors are grateful to the reviewer for the time spent and the excellent comments. All of the comments have been addressed and are included in the revised manuscript.
--------------------------------
- As advised by the reviewer, Figure 5 is included in the revised manuscript to provide pictorial presentation.
- Errors in spelling are corrected.
- Number of males and females in each group are included in Table 1.
- Correlation of BMD and vitamin C with HbA1c is included in text (lines 155-157).
- Information on LDL/HDL ratio and TG/HDL ratio in both groups is included in Table 1.
- DEXA details are corrected in Methods section.
- Discussion and conclusion includes the potential of dietary vitamin C intake in decreasing the risk of osteoporosis in our population (lines 225, 226, 234-236).

Round 2
Reviewer 1 Report
alle queries were replied. Anyway there are parts stil with DIABETIC SUBJECTS etc. Please, CORRECT with SUBJECTS WITH DIABETES